# One-shot Learning for Robot Manipulation through Egocentric Video Demonstration

## Abstract

Learning robot manipulation from egocentric video demonstrations is a challenging and promising direction for embodied intelligence, as it involves dynamic perspectives and uncertain environments. While existing methods have shown success in one-shot or few-shot learning from static videos, they are not applicable to egocentric video inputs, which significantly limits their scalability and real-world deployment. In this paper, we propose a novel coarse-to-fine directional manipulation learning framework that enables robots to acquire manipulation skills from a single egocentric video demonstration. Our method integrates an ensemble action prediction module for coarse action generation and a reinforcement learning-based refinement module for fine-grained, adaptive control. The ensemble module improves robustness by combining multiple diffusion policies, while the reinforcement module ensures accurate execution by refining motions based on real-time feedback. We evaluate our framework on three complex, multi-step manipulation tasks and demonstrate its superior performance over three state-of-the-art baselines in terms of both success rate and task robustness under one-shot egocentric settings.

## 1 Introduction

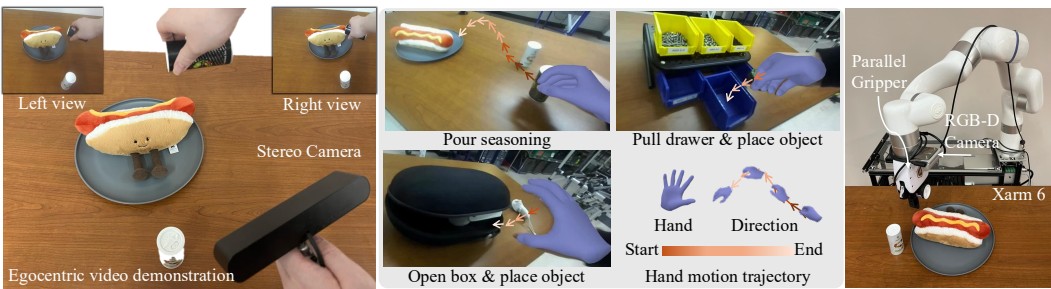

Figure 1: We propose a coarse-to-fine action prediction framework for learning robot tasks from egocentric video demonstrations. The framework requires only a single egocentric video of hand manipulation to predict both coarse- and fine-grained actions. The combined action sequence enables the robot to complete the multi-step tasks.

Robot learning from video demonstrations Kerr et al. (2024); Li et al. (2024) is considered a promising approach for effectively and efficiently acquiring manipulation skills and scaling up training datasets, as video demonstrations reduce the effort required from human demonstrators. A recent work, BiPD Zhou et al. (2025), enables one-shot learning for bimanual robot manipulation using demonstration videos and successfully performs tasks such as pulling a drawer or pouring water. This method processes the input video, extracts the trajectories of the human demonstrator's two hands, and generates actions based on these trajectories. Despite achieving one-shot learning, BiPD requires a fixed camera position and view angle, and the human hands must remain visible throughout the entire video which limits the applicability of demonstration videos.

Compared to videos with a static viewpoint, egocentric videos offer richer information about how humans perceive and interact with their environment, which is crucial for understanding human

actions. Nevertheless, learning from egocentric video demonstrations poses significant challenges, due to constantly changing camera positions and angles (see Figure 1). As a result, hand trajectories cannot be directly extracted to generate a diffusion policy as BiPD does. Additionally, for few-shot or one-shot learning, the recording perspective of egocentric videos often differs substantially from those used for task execution in real environments, which may cause the generated actions to fail. Given these challenges, the central research question is: **How can robot manipulation skills be effectively learned from only one human demonstration in an egocentric video?**

To address these challenges, we propose an ensemble-based coarse-to-fine action generation framework that efficiently learns manipulation skills from a one-shot egocentric demonstration video. Specifically, the framework consists of three modules: *3D motion extraction*, *diffusion-based coarse action prediction*, and *reinforcement learning (RL)-based action refinement*.

**3D Motion Extraction** As the human hand may not always appear in an egocentric video, it is necessary to preprocess the video and select only the frames in which a hand is visible. Subsequently, a 3D hand reconstruction from videos approach is applied to these selected frames to extract the hand motions. The resulting 3D hand motions are then used as input for the coarse action prediction.

**Diffusion-based Coarse Action Prediction** While a single diffusion policy can generate high-quality actions from video demonstrations with a static perspective, egocentric video demonstrations pose significant challenges due to constantly changing camera viewpoints and positions. Inspired by the concept of ensemble learning Rokach (2010), we propose an ensemble action prediction approach that, instead of training a single policy as in BiPD, trains multiple diffusion policies and combines them through ensembling. Additionally, we develop a contrastive orthography bagging method to assign weights to each policy based on the morphology of the interactive objects.

**Reinforcement Learning (RL)-based Action Refinement** Directly applying the predicted coarse actions to drive the robot to move may lead to failure for two main reasons: 1) The accuracy of coarse action prediction is limited due to the egocentric video input; and 2) the frame rate of coarse action prediction is low. To address this issue, we propose a reinforcement learning (RL) approach to guide the robot toward the interactive target objects identified in the egocentric video demonstration.

Unlike video-based methods that predict future frames to infer the next action Zhang et al. (2025) and approaches that rely on object meshes for pose estimation to address cross-embodiment transfer Lum et al. (2025), our method requires neither video prediction nor mesh-based pose estimation. Compared with the method that learns from egocentric video demonstrations Kareer et al. (2024), our approach enables one-shot learning, making it more applicable and efficient for real-world applications.

The contributions of this paper are summarized as follows:

• A novel coarse-to-fine action generation framework that effectively and efficiently learns robot manipulation skills from a one-shot egocentric video demonstration.

• A diffusion-based coarse action prediction method that generates coarse actions via a weighted combination of multiple diffusion policies, along with an associated contrastive morphology bagging that determines the weights based on morphology information.

• A reinforcement learning-based action refinement approach that guides robot motion toward target objects to improve action accuracy and compensate for motion drift.

## 2 RELATED WORKS

**Egocentric Video Demonstration** Different from static video demonstration, egocentric video demonstration is more challenging due to dynamic background and hand motion caused by camera motions Fan et al. (2023). Liu *et al.* Liu et al. (2022) first propose to represent the affordance in the egocentric video as an interaction heatmap using an automatic pipeline. Following that, Robo-ABC Ju et al. (2024) creates an affordance memory method based on semantic mapping and object retrieval to analyze the interaction between objects and human contact points. Some trials learn the complex manipulator behaviors by analyzing the human hand motions Grauman et al. (2024) in the video. Ego4D Grauman et al. (2022) benchmark provides hand annotations for egocentric video demonstration in 2D image space, which contains essential information about the interaction be-

tween hand and object in the video. Other information in the egocentric video demonstration, such as 3D hand pose Li et al. (2024); Kerr et al. (2024); Bahety et al. (2024), motion trajectories Chen et al. (2024); Zhang & Gienger (2024), and keypoints Gao et al. (2024); Wen et al. (2023) can be transformed into robot-related variables such as actions and trajectory. Our framework efficiently leverages a single egocentric video demonstration with hand motions to learn the manipulation task, enabling the robot to understand human actions from different perspectives.

**Trajectory Prediction** Trajectory prediction in robotics has been explored through various approaches, including reinforcement learning Ajay et al. (2023); Wang et al. (2023) and imitation learning Pearce et al. (2022); Sridhar et al. (2024). Wang *et al.* Wang et al. (2023) apply diffusion policies as an expressive policy class for offline learning. Ajay *et al.* Ajay et al. (2023) propose a conditional generative diffuser for sequential decision-making in trajectory prediction. In the imitation learning domain, ACT3D Gervet et al. (2023) and PreAct Shridhar et al. (2023) demonstrate notable improvements in low-dimensional 3D robot control. These approaches predict future trajectories based on observation-action pairs collected from expert demonstrations. Recent works Zhang et al. (2025); Lum et al. (2025) have introduced more robust trajectory prediction techniques. Yang *et al.* Yang et al. (2024) enhance generalization by integrating SIM(3)-equivariance into diffusion policies for trajectory prediction. BiPD Zhou et al. (2025) presents a bimanual robot trajectory prediction framework based on diffusion policies, incorporating action augmentation from video demonstrations. However, BiPD assumes static camera positions and fixed viewpoints in demonstration videos, and thus cannot handle egocentric video inputs. This limitation reduces its applicability, as egocentric video demonstrations both reduce the burden on human demonstrators and offer richer information about human intent and hand-object interactions. In contrast, we propose a diffusion-based method for learning skills from egocentric videos, leveraging a multi-policy ensemble for coarse action prediction and a reinforcement learning approach for action refinement.

## 3 PROBLEM SETUP

The goal of our framework is to map the robot state $S$ to the action $A$ based on the observation $O$. The action space is defined as $A = \{a^p, a^r, a^g\}$, which includes the 6-DoF pose and the binary gripper status. Since egocentric video demonstrations do not provide accurate position and orientation labels in a static world coordinate frame, actions are represented using directional instructions $d$ over a unit time period. The direction instruction $d = \{d^v, d^r\}$ consists of linear and angular velocities at each step. Egocentric video demonstrations in the observations serve as references for robot actions, where hand motions are interpreted as gripper poses from the egocentric viewpoint. Both interactive objects and hands are present in the demonstrations, and the directional instructions describe the relative movement between the hand and the interactive object.

In this paper, we consider the directional instruction at each step is driven by an ensemble action prediction and a reinforcement action refinement. The ensemble action prediction result is a combined action result. Given multiple diffusion policies $\{\pi_{\theta_i}\}_{i=1}^M$, the final action is computed by bagging their outputs as $\hat{a} = \sum_{i=1}^M w_i a_i$, where $w_i$ are normalized weights. We also leverage an off-policy reinforcement learning Haarnoja et al. (2018) to train policy $\pi$. The policy $\pi_\theta(a|s)$ is optimized to maximize the expected cumulative reward and policy entropy, formulated as: $J(\pi) = \mathbb{E}_{(s_t, a_t) \sim \rho_\pi} \left[ \sum_t r(s_t, a_t) + \alpha \mathcal{H}(\pi(\cdot|s_t)) \right]$, where $r(s_t, a_t)$ is the reward function designed to encourage fine manipulation behavior, $\mathcal{H}$ denotes the entropy of the policy, and $\alpha$ is a temperature parameter balancing reward and entropy. The following assumptions are made in our framework: 1) Egocentric video demonstrations are assumed to exhibit explicit hand-object interactions serving as informative signals for action inference. 2) The resultant action for robot manipulation can be represented as the linear combination of different actions generated from multiple diffusion policies and therefore can be composed using the parallelogram rule of vectors.

## 4 METHOD

In this section, we present the detailed architecture of the egocentric directional manipulation learning framework. An overview of the framework is illustrated in Figure 2, which consists of three main modules: the egocentric motion extraction module, the ensemble action prediction module, and the reinforcement action refinement module. The egocentric motion extraction module analyzes hand

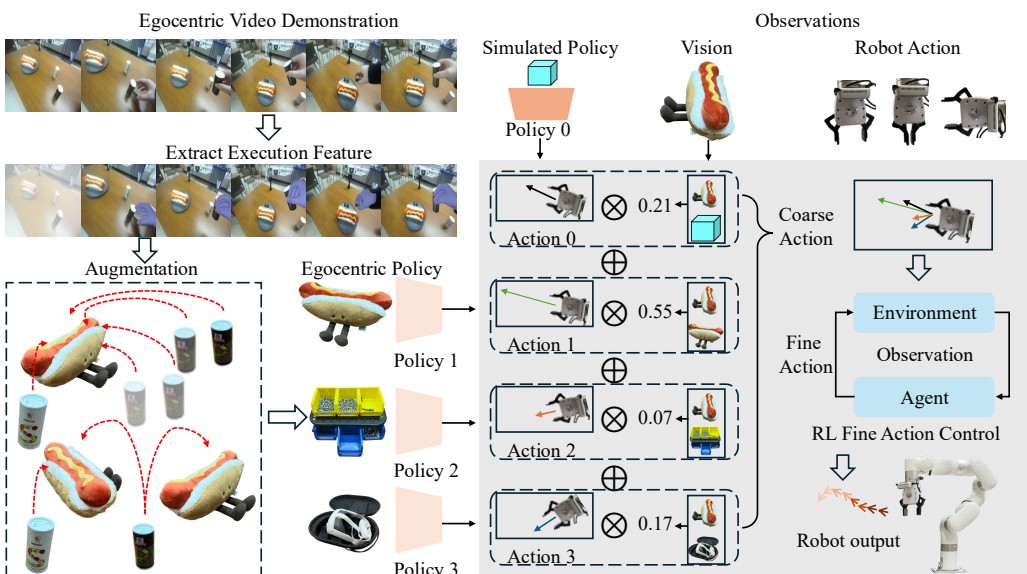

Figure 2: Overview of the coarse-to-fine action prediction framework. The input is an egocentric video demonstration. Hand execution features are extracted from the video and then transformed into robot trajectories, with frames where the hand is not visible excluded from the processing (grey color). Additional robot actions are generated through data augmentation involving positional transformations and rotations. During inference, the coarse action is combined with multiple actions and then integrated with the reinforcement fine action.

motions from the egocentric video and processes the morphology information of the interactive objects. The ensemble action prediction module employs multiple diffusion-based action models Chi et al. (2023) to predict the next action for coarse robot manipulation. Finally, the reinforcement action refinement module reduces the discrepancy between consecutive coarse action predictions by adjusting the robot's motion direction based on the egocentric video demonstration.

## 4.1 3D MOTION EXTRACTION

Hand motion in the egocentric video demonstration provides crucial information for the robot manipulation task. To collect the video demonstration, we record the egocentric videos where one hand randomly holds the camera and the other hand performs interactions with tabletop objects. In preparing an egocentric video demonstration, the recorder is required only to capture the interactive objects on the table and the manipulating hand within the field of view. This video demonstration collection setup enables a focus on hand movements while neglecting camera motion.

**Video Capturing and 3D Hand Estimation** As shown in Figure 2, the video demonstration is captured by a stereo RGB-D camera. The recorded egocentric video contains RGB frames from the left and right streams and the depth map. To extract the hand motion in the recorded egocentric video demonstration, we mainly use the left camera stream as the RGB observation and align the depth map with the left camera. Right stream frames are used for accurate 3D information Xu et al. (2023). After capturing the video demonstrations, we extract 3D hand information from the videos using WiLoR Potamias et al. (2024). Specifically, WiLoR estimates the 3D hand shape into the MANO hand representation Romero et al. (2017) and detects bounding boxes for both the left and right hands. The MANO representation is then converted into 21 joints corresponding to the hand model. While WiLoR estimates frame-level MANO representations for 3D hand poses, the resulting shapes exhibit instability due to varying camera perspectives in the egocentric video. As shown in Figure 2, the hand orientation fluctuates across frames, making the reconstructed pose unsuitable for direct use as an action reference. The hand motion extraction process is proposed to handle the perspective-changing problem in the egocentric video demonstration.

**Hand Motion Extraction** The hand motion extraction process aims to extract stabilized hand information that can be used as an action reference. The hand information $h = \{h^p, h^r, h^g\}$ includes the position, orientation, and gripping status of the hand, which are used in object-oriented hand motion trajectories. First, we remove severely fluctuating hand shapes in the egocentric video demonstration. Since only one hand typically appears in a small spatial region, cases where the detected hand switches between left and right across consecutive frames are removed. After that, we segment the interactive object in the egocentric video and compare the relative distance between the hand and the object. To identify gripper status changing time of the egocentric video demonstration, we calculate the distance between the hand motion and the interactive object. When the distance is closer than 5 millimeters, the key frame is labeled as gripper status changing time.

## 4.2 ENSEMBLE ACTION PREDICTION

Given that only one egocentric video demonstration is available for real-world action learning, the learned policy may lack robustness for real robot manipulation. To address this limitation, we introduce an ensemble action prediction module that leverages multiple policy outputs to achieve more reliable coarse action prediction. Inspired by ensemble learning Rokach (2010), we design the ensemble action prediction module based on bagging techniques, also known as bootstrap aggregating Ganaie et al. (2022). The core idea is to train multiple policies independently using a combination of simulated demonstrations and a single egocentric video demonstration and to aggregate their predicted actions through morphology bagging. Specifically, for each task, the training dataset is divided into two parts: one containing the egocentric video demonstration and the other consisting exclusively of simulated demonstrations. After generating multiple action predictions, a contrastive point cloud model Dengxiong et al. (2024) is used to evaluate the morphology of the manipulated object and to combine the actions based on morphological similarity. The final coarse control action is then derived by aggregating the outputs of multiple diffusion policies via bagging.

**Diffusion-based Action Prediction** To enhance the ability of action prediction, we propose to train different diffusion policies for action prediction. We define the base simulated training data set as $D_s = \{O_s, A_s\}$. Dataset $D_s$ is sampled from the simulation dataset from robomimc Mandlekar et al. (2023), where observation $O_s = \{o_i, s_i\}$ and action $A = \{a_i^p, a_i^r, a_i^g\}$. Then we utilize the denoising diffusion probabilistic model (DDPM) to model the conditional distribution $p(A|O)$. Starting from the noise action $a^i$, we decrease the level of noise and produce a series of actions $a_i, a_{i-1}, \cdots, a_0$ after $i$ iterations. The denoise process can be formulated as follows:

$$\mathbf{a}^{i-1} = \alpha \left( a^i - \gamma \epsilon_\theta(\mathbf{o}, a^i, i) + \mathcal{N}(0, \sigma^2, I) \right), \tag{1}$$

where $\epsilon_\theta$ is the noise prediction network. $\mathcal{N}(0, \sigma^2, I)$ is the Gaussian noise. $\alpha, \gamma$ are functions of the iteration step and can be used as learning rate scheduling in the gradient descent process Ho et al. (2020).

Diffusion models trained on simulation datasets are referred to as pre-trained diffusion policies. To adapt these models to real-world scenarios, we additionally train an egocentric diffusion policy using egocentric video demonstrations. During demonstration collection, actions $A = \{a^p, a^r, a^g\}$ are represented as relative positions with respect to the interactive object. To align these actions with the robot's coordinate frame, we transform them by adjusting their positions along the $x$, $y$, and $z$ axes. Given the limited number of egocentric demonstrations, we apply demonstration proliferation techniques, similar to those used in BiPD Zhou et al. (2025), to augment the dataset. The egocentric diffusion policy is then trained using 3D point cloud observations derived from the augmented dataset $D_e$.

**Contrastive Morphology Bagging** Although the pretrained diffusion policy and egocentric diffusion policy are prepared before the inference, the generalizability of the single diffusion model still cannot narrow the sim-to-real gap due to the lack of real-world demonstration. Inspired by the ensemble learning Rokach (2010) and the contrastive learning Chen et al. (2020), we propose a contrastive morphology bagging to determine the weight of action results generated by multiple diffusion policies. Specifically, we employ a contrastive point cloud comparison model to determine the similarity between the point cloud of objects. Similar to the point cloud similarity model Dengxiong et al. (2024), we augment the original point cloud into two samples by jittering, resizing, or flipping. Then we extract the augmented sample by the ResNet encoder $f(\cdot)$ to get the representation vectors $h_{ui}, h_{uj}$. These representation vectors are projected by a trainable MLP $g(\cdot)$ to map

the representation in the latent space. The contrastive point cloud comparison model is trained in a self-supervised manner. Given $N$ point cloud samples, the contrastive NT-Xtent loss is formulated as

$$\mathcal{L}_{i,j} = -\log \frac{\exp\left(\frac{sim(z_{ui}, z_{uj})}{\tau}\right)}{\sum_{k=1}^{2N} \mathbf{1}_{[k \neq i]} \exp\left(\frac{sim(z_{ui}, z_{uk})}{\tau}\right)}, \tag{2}$$

with the cosine similarity $sim(z_{ui}, z_{uj})$, and temperature parameter $\tau$. $\mathbf{1}_{[k \neq i]}$ is an indicator function equal to 1 only if $k \neq i$. The contrastive point cloud comparison model is able to measure the similarity between two point clouds, which can be used to compare the morphology similarity between the observed object during real-world inference and the object appearing in policy training. Therefore, we adapt the normalized similarity score between point clouds as the weight of bagging when processing multiple action outputs from diffusion policies.

### 4.3 Reinforcement Action Refinement

Since the ensemble action prediction needs to run multiple diffusion policies in one inference, the frame rate of prediction is low, and it can only be considered as a coarse result for further refinement. To refine the action for better accuracy and generalizability real-world scenarios, we propose the reinforcement action refinement to fine-grain actions based on the coarse actions. Specifically, we leverage an off-policy soft-actor-critic reinforcement learning to train policy $\pi$ and finally combine the action from contrastive morphology bagging and reinforcement policy based on the parallelogram rule of vector composition.

**Soft-Actor-Critic (SAC) Action Refinement** The SAC fine action control aims to control the end-effector move to the target keypose. We propose to use an off-policy reinforcement learning method to balance the expected rewards and the policy entropy and maximize the reward. The SAC fine action control module contains an actor network that outputs the action distribution based on the robot state and a critic network to evaluate the state-action pairs. The actor network $\pi_\theta(a|s)$ is trained by minimizing the following loss: $J_\pi(\theta) = \mathbb{E}_{s_t \sim \mathcal{D}} \left[ \mathbb{E}_{a_t \sim \pi_\theta} \left[ \alpha \log \pi_\theta(a_t|s_t) - \min(Q_{\phi_1}(s_t, a_t), Q_{\phi_2}(s_t, a_t)) \right] \right]$. The objective is to generate fine-grained actions that drive the end-effector closer to the target keypose.

**Action Composition** Based on the ensemble action prediction and reinforcement action control, the predicted actions are transformed into directional vectors. During inference, a contrastive point cloud comparison model computes a bagging score that represents the similarity between the real-world interactive object and the objects present in the diffusion policy training data. These bagging scores are used to determine the magnitude parameters of the directional vectors during action composition. The positional combination follows the parallelogram rule for vector addition, while the rotation of the end-effector is determined by the pose prediction associated with the highest bagging score. To further constrain the fine action, we define an action space around the coarse action. This action space is modeled as a sphere with a radius of 5 cm. Fine actions are executed at a higher frequency and are combined with coarse actions to ensure precise and responsive manipulation.

## 5 Experiment

The experiments aim to answer the following research questions: 1) Does the proposed coarse-to-fine action generation approach effectively generate robot actions to complete real-world manipulation tasks by learning from a one-shot egocentric demonstration? 2) Do the ensemble learning-based multi-policy composition approach and the associated contrastive morphology bagging effectively improve the success rate and overall performance of robot manipulation? 3) Does the RL-based action refinement further enhance the success rate and performance of robot manipulation?

### 5.1 Experiment Setups

The egocentric video demonstration is captured using the ZED 2 stereo camera. The camera is handheld, either in the left or right hand, allowing the user to move around the target object while completing the demonstration.

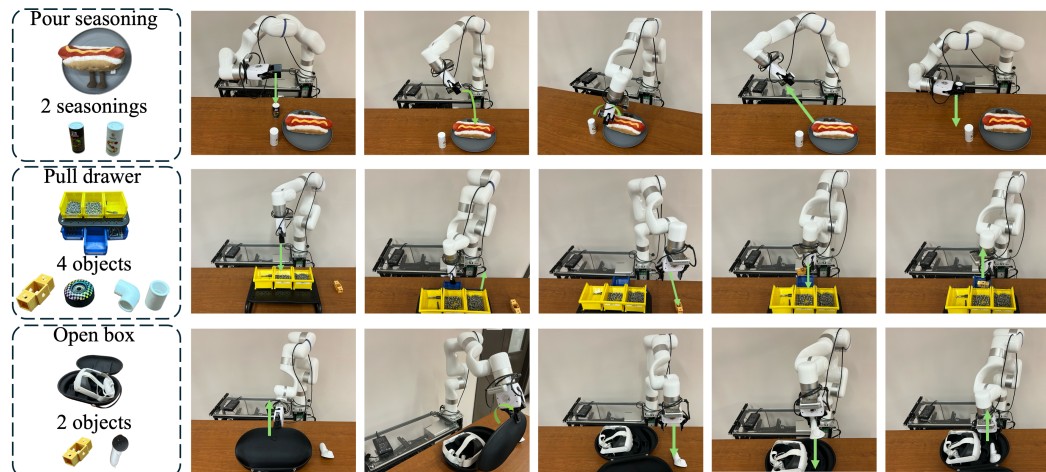

Figure 3: Tasks overview and qualitative results. Arrows indicate the intended movements.

**Task Description** We evaluate our framework on three real-world multi-step tasks: *Pull Drawer*, *Open Box*, and *Pour Seasoning*. These tasks require the robot to perform basic manipulation actions and interact with objects. *Pull Drawer*: The robot pulls out the drawer beneath the shelf, picks up a wheel or a 3D-printed structure, and places the object back into the drawer. *Open Box*: The robot first holds the case, rotates the gripper to open it, and then picks up a controller or a 3D-printed structure from the table. *Pour Seasoning*: The robot picks up a seasoning container from the table, moves to the plate, and pours the seasoning onto it.

**Evaluation** We evaluate our framework on three tasks learned from egocentric video demonstrations, using two metrics: average task length and final success rate. The average length follows the definition used in CLAVIN Mees et al. (2022), where each substep is considered as part of a sequential long-horizon task. The final success rate refers to the success rate of the last substep. For the *Pull Drawer* task, we evaluate the framework using four different objects, with 10 trials per object. For both the *Open Box* and *Pour Seasoning* tasks, we conduct 20 trials across two different objects. In addition, we report the number of successful trials for each subtask (e.g., Reach, Open, Release).

## 5.2 RESULTS

| **Open Case Tasks** | | | | | | | |
|---|---|---|---|---|---|---|---|
| **Process** | Reach | Open | Release | Grab | Place | Ave. Len. | Succ. Rate |
| ACT Zhao et al. (2023) | 36 | 20 | 18 | 5 | 3 | 2.05 | 0.075 |
| DP3 Ze et al. (2024) | 39 | 28 | 18 | 12 | 7 | 2.60 | 0.175 |
| Ours | 40 | 34 | 33 | 27 | 22 | 3.90 | 0.550 |
| **Pour Seasoning Tasks** | | | | | | | |
| **Process** | Grab | Move | Pour | Place | – | Ave. Len. | Succ. Rate |
| ACT Zhao et al. (2023) | 37 | 30 | 6 | 2 | – | 1.875 | 0.050 |
| DP3 Ze et al. (2024) | 35 | 24 | 16 | 8 | – | 2.075 | 0.200 |
| Ours | 40 | 35 | 35 | 33 | – | 3.575 | 0.825 |
| **Pull Drawer Tasks** | | | | | | | |
| **Process** | Reach | Pull | Leave | Grab | Place | Ave. Len. | Succ. Rate |
| ACT Zhao et al. (2023) | 35 | 13 | 8 | 5 | 2 | 1.575 | 0.050 |
| DP3 Ze et al. (2024) | 38 | 30 | 28 | 18 | 9 | 3.075 | 0.225 |
| BiDP Zhou et al. (2025) | 40 | 30 | 30 | 26 | 22 | 3.700 | 0.550 |
| Ours | 40 | 33 | 32 | 27 | 23 | 3.875 | 0.575 |

Table 1: Performance of open case, pour seasoning, and pull drawer. Each task consists of 40 independent trials.The number of trials successfully accomplished in each subtask is shown, highlighting differences in task complexity and final success rates across scenarios.

| Open Case Tasks | | | | | | | |
|---|---|---|---|---|---|---|---|
| **Process** | Reach | Open | Release | Grab | Place | Ave. Len. | Succ. Rate |
| Classical control | 36 | 29 | 21 | 20 | 20 | 3.15 | 0.50 |
| Behavior cloning | 39 | 33 | 30 | 24 | 21 | 3.675 | 0.525 |
| Ours | 40 | 34 | 33 | 27 | 22 | 3.90 | 0.550 |
| **Pour Seasoning Tasks** | | | | | | | |
| **Process** | Grab | Move | Pour | Place | – | Ave. Len. | Succ. Rate |
| Classical control | 39 | 35 | 33 | 31 | - | 3.45 | 0.775 |
| Behavior cloning | 40 | 34 | 32 | 30 | - | 3.40 | 0.75 |
| Ours | 40 | 35 | 35 | 33 | - | 3.575 | 0.825 |
| **Pull Drawer Tasks** | | | | | | | |
| **Process** | Reach | Pull | Leave | Grab | Place | Ave. Len. | Succ. Rate |
| Classical control | 37 | 29 | 22 | 21 | 20 | 3.325 | 0.50 |
| Behavior cloning | 40 | 33 | 30 | 23 | 20 | 3.65 | 0.50 |
| Ours | 40 | 33 | 32 | 27 | 23 | 3.875 | 0.55 |

Table 2: Comparison of classical control Coleman et al. (2014), behavior cloning control Torabi et al. (2018), and our proposed solution. We report the number of successful trials for each subtask, the average completion length, and the final task success rate.

We compare our framework with two teleoperation demonstration-based methods and one video demonstration-based learning method. **Action Chunking Transformers (ACT)** Zhao et al. (2023) is a transformer-based model that learns from RGB frames and teleoperated demonstration data. **3D Diffusion Policy (DP3)** Ze et al. (2024) is a diffusion policy model that uses a point cloud encoder. **BiDP** Zhou et al. (2025) is a diffusion policy that learns from a single static video demonstration to perform dual-arm manipulation. The *Open Box* and *Pour Seasoning* tasks are evaluated using ACT and DP3 but not BiDP, as BiDP is not applicable to these tasks. The *Pull Drawer* task is evaluated using all three baselines.

The coarse-to-fine action generation framework effectively enhances action prediction capabilities (**Q1**). With the ensemble action prediction module, the coarse action prediction successfully accomplishes multi-step tasks with a longer average length, indicating greater robustness at each subtask compared to ACT, DP3, and BiDP. As shown in Table 1, the reinforcement-based action refinement effectively mitigates action drift between two consecutive ensemble predictions, guiding the end-effector accurately toward the interactive target and improving the final success rate. In all three tasks, our framework outperforms all compared baselines. In the *Pour Seasoning* task, our framework achieves a success rate exceeding 82%. We also compare our method with BiPD on a similar *Pull Drawer* task, which requires pulling out the drawer, picking up an object, and placing it back into the drawer (Table 1). Our framework demonstrates superior performance in handling each subtask and completing the overall multi-step tasks from a single egocentric video demonstration.

We further compare our proposed reinforcement fine-control method with behavior cloning control Torabi et al. (2018), and classical control Coleman et al. (2014). As shown in Table 2, behavior cloning is stable when the action sequence is complex but highly imitable, achieving competitive results. However, it generalizes poorly when the scene varies; in the more variable Pour Seasoning task its success rate drops to 0.75, trailing both the feedback controller and our method. The feedback controller can exploit visual–spatial cues more accurately and is relatively tolerant to scene changes, but it accumulates trajectory bias during imitation—reflected in lower success and shorter average completion length. In contrast, our SAC consistently achieves the best overall performance: the highest success rates across all three tasks, and the longest average completion lengths. From Table 2, the classical control baselines show a sharp drop in success on the open-case (open-and-pull) and pull-drawer tasks. Both are contact-rich and require fine end-effector adjustments on end effector, which is difficult for the classical control on robot. Behavior cloning often drifts in the terminal phase (e.g., approach angle predicted in the grab task), reducing reliability, whereas our approach remains robust across these variations. These results indicate that starting from demonstrations and then improving the policy with reinforcement learning enables SAC to master complex action chains and generalize better to scene variation, mitigating behavior cloning's covariate-shift issues and outperforming the geometric alignment ability of the feedback controller.

| Point Cloud | Bagging | Key frames | Action Space | Reach | Pull | Leave | Grab | Place | Ave. Len. | Succ. Rate |
|---|---|---|---|---|---|---|---|---|---|---|
| ✗ | ✗ | ✗ | ✗ | 36 | 23 | 22 | 18 | 9 | 2.70 | 0.225 |
| ✓ | ✗ | ✗ | ✗ | 38 | 26 | 22 | 15 | 13 | 2.85 | 0.325 |
| ✓ | ✓ | ✗ | ✗ | 40 | 28 | 25 | 23 | 19 | 3.375 | 0.475 |
| ✓ | ✓ | ✓ | ✗ | 40 | 31 | 29 | 26 | 22 | 3.70 | 0.550 |
| ✓ | ✓ | ✓ | ✓ | 40 | 32 | 32 | 27 | 23 | 3.85 | 0.575 |

Table 3: Ablation study showing the effect of point cloud, bagging, keyframe supervision, and action space constraint. ✓: enabled, ✗: disabled.

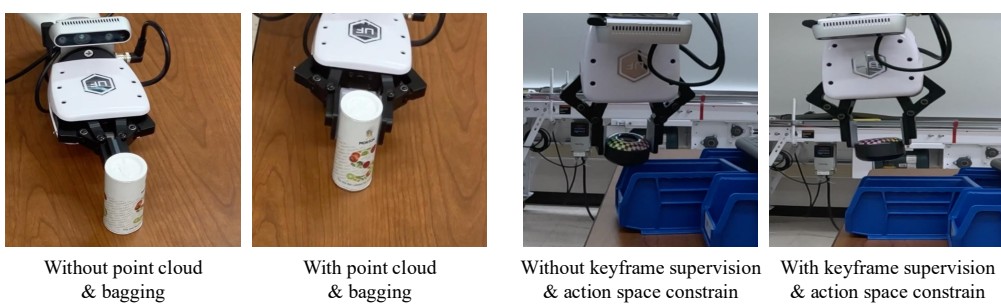

Without point cloud & bagging     With point cloud & bagging     Without keyframe supervision & action space constrain     With keyframe supervision & action space constrain

Figure 4: Qualitative results for ablation study on proposed module. The proposed modules generate action predictions that better adapt to the environment.

We also conduct an ablation study to evaluate the contributions of each component in our framework, demonstrating their effectiveness in improving robot manipulation performance (**Q2, Q3**). As shown in Table 3, *Point Cloud* refers to using point cloud data as input to the diffusion policy. *Bagging* indicates the coarse action combination based on contrastive morphology bagging. *Key Frames* represents whether the relative distance between hand motion and the object is used to determine the gripper status. *Action Space* denotes whether action space constraints are applied during reinforcement-based fine control. The results show that the bagging, 3D hand motion extraction, and reinforcement learning-based action refinement significantly improve both the average number of completed subtasks and the final success rate.

We also provide some qualitative results in the ablation study. In the seasoning-pouring task (Figure 4), with point-cloud and bagging assistance, the gripper securely grasps the salt bottle; without these modules, it fails to maintain a stable grasp due to the lack of morphological cues. As shown in Figure 4, with the keyframe supervision and action space constrain, the robot can move to a better pose to place the wheel in the drawer. The keyframe supervision and action space constrain helps hanging the wheel above the drawer. This allows the robot to decrease the risk of place the wheel outside the drawer.

## 6 CONCLUSION

In this work, we propose a novel coarse-to-fine action generation framework that enables robots to learn manipulation skills from a single egocentric video demonstration. By integrating 3D hand motion extraction, an ensemble-based diffusion policy for coarse action prediction, and a reinforcement learning-based action refinement, our method effectively addresses the challenges posed by egocentric viewpoints and one-shot learning. Extensive evaluations on three real-world multi-step tasks demonstrate that our approach significantly outperforms state-of-the-art baselines in both task success rate and robustness. Additionally, the ablation study confirms the importance of each component in improving performance. The proposed modules show better performance in handling manipulation details by fully using the additional information provided in the egocentric video demonstration. Our framework opens up new possibilities for scalable and efficient robot learning from unstructured, first-person video data, making it well-suited for practical deployment in real-world environments.

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

## A  APPENDIX

**Problem Definition** The demonstrations are captured as egocentric videos with a non-static camera perspective that naturally follows the user's hand during manipulation. The task we define is a one-shot learning task, as our setting provides only a single real demonstration per test task, and the experimental task is not identical to the pretraining data. Since the dynamic viewpoint may not consistently capture both the hand and the object in every frame, our framework models the hand's motion trajectory as the primary signal, using object observations as reference cues only when they are visible.

**Simulation Data** To equip our policy with basic manipulation primitives, we pre-train it on simulated pick-and-place trajectories derived from common daily activities. These include reaching, grasping, and placing a variety of simple objects (e.g., controllers, boxes) in MuJoCo, without any particular selection to closely match the testing tasks. The simulated trajectories provide diverse motion examples that can be reused across tasks, effectively bootstrapping the policy prior to fine-tuning on the single available real-world demonstration.

**Object and Morphology Comparison** For each task we first segment the "interactive region" in the wrist-camera RGB-D stream, for example, the box and its contents in the "open box" task, by generating a binary mask of pixels corresponding to the object(s) of interest. Segmentation results assist the model to focus on the interactive object in the RGB image and the depth map. We then extract two parallel feature streams: (1) RGB branch: a standard 3-channel ResNet backbone (2) Depth branch: a single-channel ResNet backbone Their outputs are concatenated and passed through a lightweight fusion layer that learns per-modality weights. The resulting fused feature vector encodes the object's morphology, which we use to compare against each policy's reference morphology during ensemble-weight computation. We estimate the hand–object distance directly in the camera coordinate frame by projecting both the segmented hand and object point clouds from the same RGB-D image using the camera's intrinsic parameters. Because the hand and object are observed under identical lighting and viewpoint, this relative distance remains consistent and reliable. We extract "key frames" from the egocentric video only when the entire hand is fully visible, and we segment both the hand and object using the Segment Anything Model (SAM) to ensure precise mask generation before computing their spatial relationship.

**Ensemble Weight** At every coarse-level decision step, we update the ensemble weights by measuring the similarity between the current object morphology and each policy's stored morphology prototype. The policy's stored morphology prototype is collected from the one-shot egocentric video demonstration. 1. Observation: The wrist camera provides an RGB image and a depth map. 2. Feature extraction and fusion: As described above, these inputs are processed to yield a morphology embedding $f$. 3. Similarity: We compute the cosine similarity between the policy's stored object feature $f_p$ and the observed object feature $f_o$, obtained after projection from the raw input. 4. Normalization: We apply a softmax over these cosine scores to produce positive, sum-to-one weights $\{w_i\}$. These weights are recomputed at each coarse-level step based on the latest observation, allowing the ensemble to dynamically adapt to changes in object appearance and pose.

**RL Details** The fine-level control will apply after determine the ensemble weights. The frequency ratio between coarse-level control (Ensemble Action Prediction) and fine-level control (Reinforcement Action Refinement) is 1:10. We define the reward $r_t$ as: $r_t = -\alpha||p_t - p_{\text{target}}||_2 + k\mathbf{1}(\Delta\theta_t \leq \theta_{\max}) + \beta\mathbf{1}(d_m \leq d_c \ \wedge \ \Delta\theta_t \leq \theta_{\max}) - \delta\mathbf{1}(\Delta\theta_t > \theta_{\max})$, where the function represents the distance reward and the action space reward for fine-level control. The coefficient $\alpha, k, \beta, \delta$ can adjust the reward. The robot gets a higher reward when end-effector $p_t$ is close to the target position $p_{target}$. The action space reward is based on the direction of the next position and the moving distance $d_m$ of each step. The moving distance threshold of each step is $d_c$, and the direction angle threshold is $\theta_{max}$. The robot is encouraged to move forward in the action space aligned with the coarse-level direction and receives a smaller penalty when the end-effector is closer to the target.

**Baseline Comparison** The pull-drawer task in our setup closely mirrors the drawer-opening behavior used in BiDP, whereas the other two tasks involve actions that differ substantially from those in BiDP. To ensure a fair comparison, all baseline methods were trained exclusively on the demonstrations recorded for this study.

