# OpenReview forum: "One-shot Learning for Robot Manipulation through Egocentric Video Demonstration"
_ICLR.cc/2026/Conference — ICLR 2026 Conference Withdrawn Submission_

### Official Review · Reviewer_a9Ht · 2025-10-29

**Soundness:** 2
**Presentation:** 2
**Contribution:** 2
**Rating:** 2
**Confidence:** 4

**Summary:**

This paper proposes a novel "coarse-to-fine" framework enabling a robot to learn complex manipulation skills from a single egocentric video demonstration. The authors aim to solve the challenges posed by dynamic camera perspectives in egocentric videos, which prevent the use of existing one-shot learning methods designed for static cameras. The framework consists of three main modules: a 3D motion extraction module to process hand movements from the video, an ensemble action prediction module that uses multiple diffusion policies (weighted by "contrastive morphology bagging") to generate a "coarse" action, and a reinforcement learning (RL) module to "refine" this action for precise, adaptive control. The method was evaluated on three multi-step tasks (Pull Drawer, Open Box, and Pour Seasoning), where it reportedly outperformed three state-of-the-art baselines in both success rate and robustness. An ablation study also confirms that each component positively contributes to the system's performance.

**Strengths:**

- This paper is claimed to be one of the first to enable one-shot learning from a single *egocentric* video, which is very challenging due to dynamic camera views. The problem is difficult and important.

- The method intelligently combines an ensemble of diffusion policies for a robust "coarse" action with a reinforcement learning module for "fine-tuning" and error correction.

- The method beat three other state-of-the-art methods in success rates on three complex, real-world robot tasks

**Weaknesses:**

- The "one-shot" claim is misleading. The method relies heavily on pre-trained policies from a large simulation dataset ($D_{s}$) and a separate pre-trained point cloud model. The single egocentric video is only used for *adaptation*, not learning from scratch.

- The paper admits that its initial 3D hand pose estimates from WiLoR "exhibit instability" and "fluctuate across frames". Its solution is to "remove severely fluctuating hand shapes" and "cases where the detected hand switches". These are vague heuristics, not a robust method. It's unclear how much data is discarded or how this "cleaning" process affects the final demonstration quality.

- The "contrastive morphology bagging" weights policies based on the visual similarity of objects. The paper fails to justify why visual morphology is the correct or best proxy for weighting action policies. A policy trained on a visually similar object is not guaranteed to provide a better action if the required interaction (e.g., stiffness, weight, articulation) is different.

- The experiments are missing the most critical ablation: performance *without* the simulation pre-training. This makes it impossible to know if the "one-shot" video provides any significant learning or just minor tuning.

- Another key ablation is missing: the paper's complex "contrastive morphology bagging" is never compared to a simple, unweighted average of the ensemble policies. This fails to prove the bagging component is necessary.

- The paper completely ignores the high cost of its "single demonstration." It requires a stereo RGB-D camera, complex 3D hand reconstruction, and object segmentation (which may even need much human effort). This may be far more effort than collecting more, simpler teleoperation data.

- The RL refinement module is trained to chase a "target keypose" ($p_{target}$), but the paper never explains where this critical target position comes from.

- The baseline comparisons may be unfair. The paper notes that methods like BiPD require a static camera and are thus being tested on a problem (egocentric video) they were not designed to solve.

**Questions:**

See the weakness section

---

### Official Review · Reviewer_3MgG · 2025-10-31

**Soundness:** 2
**Presentation:** 2
**Contribution:** 3
**Rating:** 4
**Confidence:** 3

**Summary:**

The paper addresses one-shot imitation learning for robot manipulation from a single egocentric video, a setting that is more challenging than learning from static videos due to dynamically changing viewpoints. To tackle this, the authors propose three modules: an egocentric motion-extraction module, a diffusion-based ensemble action-prediction module, and a Soft Actor-Critic (SAC)–based action-refinement module. Real-world experiments on three manipulation tasks demonstrate improved performance over three state-of-the-art baselines.

**Strengths:**

The paper tackles a practical and widely applicable problem: one-shot learning from an egocentric demonstration. It thoughtfully combines established components—3D hand estimation, diffusion-based policies, point-cloud augmentation, and SAC-based policy refinement. In particular, the idea of ensemble action prediction using multiple diffusion policies pretrained in simulation, together with contrastive morphology bagging, appears novel in this context. The overall approach achieves superior performance on three robot manipulation tasks compared with existing methods.

**Weaknesses:**

The method relies on a strong assumption that actions can be represented as a linear combination of the pretrained diffusion policies, which may limit applicability when test environments differ substantially from the simulation domain used for pretraining. Because the method depends on simulation data, it may be more accurate to frame it as “one-shot sim-to-real transfer learning,” to avoid confusion that the approach learns solely from a single real video. While the results are promising, aspects of the experimental setup are under-specified. For example, it is unclear whether the baselines also use simulation data. A detailed analysis of distributional differences among (a) the simulation videos used for pretraining, (b) the one-shot demonstration, and (c) the evaluation setting—covering interactive object types, background scenes, and camera viewpoints—would make the results more convincing. Finally, the paper lacks several details in the methods and the experiments section, which are specified in the questions below.

**Questions:**

- What is the rationale for the linear-combination assumption? In a one-shot setting, aren't test-time tasks expected to be novel relative to the simulation data?

- How different are the simulated data, one-shot demonstration, and the evaluation setting (e.g., object categories, backgrounds, camera motion/viewpoints)?

- Do the baselines use the same amount and type of data, including any simulation data for diffusion-policy pretraining?

- Which task(s) were used in the ablation study reported in Table 3? Also, which components are the key to making the approach applicable to egocentric videos? For example, if we apply all the techniques that can also be applied to the baselines, does the proposed approach still outperform the baselines?

- Why is BiPD applicable to Pull Drawer but not to the other two tasks?

- How is the interactive object segmented during hand-motion extraction? Does this rely on an external segmentation model expected to generalize across objects and scenes?

- Is an RGB-D camera necessary? Could comparable 3D information be obtained from stereo images only, as in BiPD?

- Can the framework be naturally extended to few-shot learning? If so, what modifications would be required?

---

### Official Review · Reviewer_7ZX7 · 2025-10-31

**Soundness:** 2
**Presentation:** 2
**Contribution:** 3
**Rating:** 4
**Confidence:** 4

**Summary:**

This paper proposes a coarse-to-fine egocentric manipulation learning framework that enables robots to learn manipulation skills from a single egocentric video demonstration. The system integrates three key components: a 3D motion extraction module for hand and object tracking, an ensemble action prediction module combining multiple diffusion policies via morphology-aware bagging, and a reinforcement learning (SAC) module for fine-grained action refinement. Experiments on three real-world multi-step tasks show that this approach outperforms existing baselines such as ACT, DP3, and BiPD.

**Strengths:**

1. The study focuses on egocentric video demonstrations, which provide richer perception of human intent and are more natural for large-scale data collection than static third-person views.
2. The integration of ensemble diffusion policies with morphology-based weighting introduces a practical method to enhance robustness from limited data, addressing one-shot learning challenges.
3. The coarse-to-fine refinement design, combining diffusion-based prediction with RL fine-tuning, is conceptually sound and experimentally validated through multiple tasks and ablation studies.

**Weaknesses:**

1. Despite claiming “real-world” validation, the experimental setting is still limited and constrained—tasks are simple, tabletop, and lack environmental variability or deformable-object interactions; no generalization tests across scenes or objects are reported.
2. The egocentric motion extraction module is described ambiguously—key technical details such as noise filtering, frame selection, or 3D hand–object distance computation are underexplained, reducing reproducibility.
3. The system’s computational and inference complexity may be excessive: running multiple diffusion policies plus an RL controller could limit real-time deployment, but no runtime or resource analysis is provided.
4. The writing lacks important experimental clarity—it does not specify how much data is used to train each diffusion policy, how augmentation expands one-shot data, or what data sources are used for baselines, which weakens methodological transparency.

**Questions:**

1. How scalable is the approach when applied to new tasks—does it require recording a new egocentric video per task?

2. Could the authors quantify the computational cost (GPU memory, inference latency) introduced by the ensemble diffusion module?

3. How robust is the system to errors in 3D hand reconstruction or missing frames during motion extraction?

4. Can the framework generalize to more diverse manipulations (e.g., deformable or multi-object tasks) beyond tabletop scenes?

---

### Official Review · Reviewer_7xoQ · 2025-10-31

**Soundness:** 2
**Presentation:** 2
**Contribution:** 2
**Rating:** 2
**Confidence:** 4

**Summary:**

This paper presents a framework that enables robot to learn manipulaiton task from a single egocentric video demonstration. This paper first utilizes a 3D motion extraction module to extract coarse action from human video. Further, this framework train multiple diffusion policy to ensumble coarse action. In the end, a reinforcement-learning based method is utilized for action refinement. The results prove the effectiveness of this method.

**Strengths:**

1. This paper proposes a framework to learn from human video.
2. This paper provides the real-world video to prove the effectiveness of this method.

**Weaknesses:**

1. It is unclear why this framework needs to train multiple diffusion policies and ensemble these actions. I think the more more conventional and seemingly advantageous approach would be to train a single, unified diffusion policy on the aggregated dataset of all human demonstrations, which could maybe enhance the perception ability of the shared pointcould encoder. However, I could not understand why this method could work. It is strange that an open bag policy could provide weight for pick object task.

2. The experimental evaluation appears to be limited, as the proposed method is primarily compared against basic baselines. The evaluation omits crucial comparisons to several state-of-the-art methods in the field of robotic manipulation from video demonstration[1].

[1] You Only Teach Once: Learn One-Shot Bimanual Robotic Manipulation from Video Demonstrations

**Questions:**

1. What is the difference between this paper and the BiPD paper[1]? It seems the whole pipeline is the same sa the BiPD paper, and only utilizes an RL-based action refinement.



[1] You Only Teach Once: Learn One-Shot Bimanual Robotic Manipulation from Video Demonstrations

---

### Note · Authors · 2025-11-26

**Comment:**

I will revise this paper based on the review comments and submit to another conference.

**Withdrawal Confirmation:**

I have read and agree with the venue's withdrawal policy on behalf of myself and my co-authors.